# An Efficient Multi-Level Convolutional Neural Network Approach for White Blood Cells Classification

**DOI:** 10.3390/diagnostics12020248

**Published:** 2022-01-20

**Authors:** César Cheuque, Marvin Querales, Roberto León, Rodrigo Salas, Romina Torres

**Affiliations:** 1Facultad de Ingeniería, Universidad Andres Bello, Viña del Mar 2531015, Chile; c.cheuquecerda@uandresbello.edu (C.C.); roberto.leon@unab.cl (R.L.); 2Escuela de Tecnología Médica, Universidad de Valparaíso, Viña del Mar 2540064, Chile; marvin.querales@uv.cl; 3Centro de Investigación y Desarrollo en Ingeniería en Salud, Escuela de Ingeniería C. Biomédica, Universidad de Valparaíso, Valparaíso 2362905, Chile; rodrigo.salas@uv.cl; 4Instituto Milenio Intelligent Healthcare Engineering, Valparaíso 2362905, Chile

**Keywords:** white blood cells classification, deep learning, multi-level classification, multi-source datasets

## Abstract

The evaluation of white blood cells is essential to assess the quality of the human immune system; however, the assessment of the blood smear depends on the pathologist’s expertise. Most machine learning tools make a one-level classification for white blood cell classification. This work presents a two-stage hybrid multi-level scheme that efficiently classifies four cell groups: lymphocytes and monocytes (mononuclear) and segmented neutrophils and eosinophils (polymorphonuclear). At the first level, a Faster R-CNN network is applied for the identification of the region of interest of white blood cells, together with the separation of mononuclear cells from polymorphonuclear cells. Once separated, two parallel convolutional neural networks with the MobileNet structure are used to recognize the subclasses in the second level. The results obtained using Monte Carlo cross-validation show that the proposed model has a performance metric of around 98.4% (accuracy, recall, precision, and F1-score). The proposed model represents a good alternative for computer-aided diagnosis (CAD) tools for supporting the pathologist in the clinical laboratory in assessing white blood cells from blood smear images.

## 1. Introduction

The peripheral blood smear represents a routine laboratory test that provides the physician with a great deal of information about a patient’s general condition. It provides a qualitative and quantitative assessment of blood components, mainly cells and platelets. Blood cells can be divided into white blood cells (WBC) or leukocytes and red blood cells (RBC) or erythrocytes [1]. Leukocytes, in turn, comprise five types of nucleated cells: *monocytes, basophils, eosinophils, neutrophils,* and *lymphocytes*. The total WBC count as the difference of percentage between the subtypes provides critical information in infectious diseases and chronic processes such as anemia, leukemia, and malnutrition [2].

The manual WBC count is based on the microscopic observation of the blood smear by the analyst, who can differentiate the subtypes mainly based on the morphological characteristics of the cell nucleus and cytosol. However, this process strongly depends on the analyst’s time and experience, leading to errors if the analyst is not adequately trained [3]. In addition, since this hematological evaluation is a routine test, it is often in high demand in clinical laboratories, representing an increased workload that affects performance. Thus, providing computer-aided diagnosis (CAD) tools for diagnostic assistance in the laboratory is required. For instance, CAD systems have been developed that use image processing techniques to classify the differential white blood cell count [4,5]. This automatic leukocyte classification allows a faster and more reproducible result to be generated while reducing bias and inter-observer variability.

On the other hand, due to the inherent complexity of observing medical examinations, their automation is still a challenge for researchers today. Achieving levels of precision comparable to those of a professional in the area is critical. The automatization is prone to errors or biases in recognizing and classifying these images. These will directly impact the diagnosis, increasing the treatment costs and negatively affecting the recovery and survival of patients.

The main component of the computer-aided system for WBC classification is the cell detection and segmentation algorithm. Based on image processing analysis, it will identify the different elements of interest considering various aspects associated with cell morphology: size, shape, texture, nucleus, etc. [4]. Commonly, cell segmentation is a complex task in tissue samples. However, this task is straightforward in cell smears, given the dark nucleus in leukocyte staining. The challenge is mainly in delineating cell borders, separating overlapping cells, and removing noise and artifacts in the image during acquisition [6]. Given the advantages of artificial intelligence in image processing, several machine learning (ML) alternatives have been evaluated to classify and segment leukocytes. These methods range from the support vector machine (SVM) [7,8] and Naïve Bayesian [9,10] to more complex algorithms such as deep learning (DL) models [11,12,13].

Within DL models, convolutional neural networks (CNNs) have shown exemplary performance in medical image classification [14,15,16,17]. CNNs are feed-forward neural networks that can be divided into two main parts: deep convolutional feature extraction and classification. This model extracts features by applying multiple convolutional and pooling layers that include linear operations (called kernels) that emphasize an input image’s characteristic. A fully connected dense layer does the classification step to learn the model using the extracted features [18,19]. Following this structure, several types of CNN models have been proposed for specific tasks such as classification, among which are AlexNet [20], ResNet [21], VGG [22], and GoogLeNet [23], and segmentation, highlighting the fully connected network (FCN) [24], U-Net [25], and Faster-RCNN [26], these being applied in the processing of blood smear images for differential WBC counting, achieving good performance results. Recently, an efficient network architecture called MobileNet was proposed as a small, lightweight, and low latency model for mobile and embedded vision applications [27]. This model has been demonstrated to be effective when applied to various tasks.

Most researchers have used these methods as one-level designs or single models built on the entire dataset. In other fields, these single models may experience difficulty in handling the increasingly complex data distribution [28] or large-scale visual recognition [29]. Given the characteristics of leukocytes, better performance in WBC classification could be obtained if a multi-level scheme is developed. In the first level of this scheme, the polymorphonuclears are separated from the mononuclear. Afterward, on the one hand, the monocytes and lymphocytes (mononuclear) and, on the other hand, the neutrophils, eosinophils, and basophils (polymorphonuclears) are classified in the second level. Thus, more features could be extracted from each cell image, and the classification performance could be increased.

On the other hand, medical datasets acquired from several institutions may have different quality, contrast, and acquisition mechanisms. Thus, the datasets are prone to an inherent bias caused by various confounding factors [30]. In machine learning, the dataset bias may lead to a difference between the estimated and the true value of desired model parameters. A possible solution to this problem is combining multi-source datasets such that the model will be robust to the unseen domains with better generalization performance.

Therefore the main objective of this research is to develop a multi-level convolutional neural network (ML-CNN) model to improve the automatic detection and classification of individual white blood cells by mitigating the dataset bias with the combination of multi-source datasets. The main contribution of this work is twofold. First, a new multi-level deep learning algorithm separates the leukocyte detection and classification processes into two levels. In the first level, the Faster R-CNN network is applied to identify the region of interest of white blood cells, together with the separation of mononuclear cells from polymorphonuclear cells. Once separated, two parallel CNN models are used to classify the subclasses in the second level. Second, the ML-CNN proposal was implemented using the MobileNet architecture as the base model. It is an efficient model with an adequate balance between high performance and structural complexity, making its implementation in automated equipment such as a CAD system feasible. Furthermore, the MobileNet applies the depthwise separable convolution to extract relevant features from each channel, which better uses the information contained in the images to improve leukocyte classification.

The article is structured as follows. In Section 2, we present a brief revision of the state of the art. In Section 3, the proposed method is presented. Results and discussion are presented in Section 4. Finally, in Section 5, we give some concluding remarks, and we outline some future works.

## 2. State of the Art

Traditional machine learning (ML) and deep learning (DL) models have been extensively proposed as alternatives for the automatic classification of leukocytes [5,31,32]. Such is the case of Abou et al. [33], who developed a CNN model to identify WBC. Likewise, Togacar et al. [34] proposed a subclass separation of WBC images using the AlexNet model. In addition, Hegde et al. [13] proposed a deep learning approach for the classification of white blood cells in peripheral blood smear images. Wang et al. [35] applied deep convolution networks on microscopy hyperspectral images to learn spectral and spatial features. This form makes full use of three-dimensional hyperspectral data for WBC classification. Basnet et al. [36] optimized the WBC CNN classification enhancing loss function with regularization and weighted loss, decreasing time processing. Jiang et al. [37] constructed a new CNN model called WBCNet that can fully extract features of the WBC image by combining the batch normalization algorithm, residual convolution architecture, and improved activation function. Other authors include steps to improve the feature extraction process. Thus, Yao et al. [38] proposed the two-module weighted optimized deformable convolutional neural networks (TWO-DCNN) white blood classification, characterized as two-module transfer learning and deformable convolutional layers for the betterment of robustness.

A novel blood-cell classification framework named MGCNN, which combines a modulated Gabor wavelet with CNN kernels, was proposed by Huang et al. [39]. In this model, multi-scale and orientation Gabor operators are taken dot product with initial CNN kernels for each convolutional layer. Experimental results showed that MGCNN outperformed traditional SVM and single CNN networks. Likewise, Khan et al. [40] presented a new model called MLANet-FS, which combined an AlexNet network with a feature selection strategy for WBC-type identification. Razzak et al. [41] proposed a WBC segmentation and classification using a deep contour aware CNN and extreme machine learning (ELM).

Another interesting strategy is to apply hybrid methods such as the proposal of Çınar and Tuncer [23]. They presented an approach combining AlexNet and GoogleNet to extract features from WBC images. Then, these features are concatenated and classified using SVM. In the same way, Özyurt [42] proposed a fused CNN model for WBC detection where the pre-trained architectures AlexNet, VGG-16, GoogleNet, and ResNet were used as feature extractors. Then, the features obtained were combined and later classifier using ELM. Patil et al. [43] presented a deep hybrid model, CNN and recurrent neural networks (RNN), with canonical correlation analysis to extract overlapping and multiple nuclei patches from blood cell images. As a novel proposal, Baydilli and Atila [44], to overcome the problem of data sets, developed a leukocyte classification via capsule networks, an enhanced deep learning approach. A capsule network consists of the decoder (used to reconstruct the image) and the encoder (responsible for extracting features and classifying the image).

Research presenting a multi-level scheme in the WBC classification is scarce. Although Baghel et al. [45] do not perform a plan, they propose a CNN classification model whose performance is evaluated in two phases: an initial step, binary discrimination between the mononuclear and polymorphonuclears, and a second phase that corresponds to the classification of subtypes. Tran et al. [46] presented as an initial setup a DL semantic segmentation between WBC and RBC, as initial steps for classifying leukocytes. However, none of these methods has a similar scheme to the one proposed in our research, so we consider that our algorithm is novel and can efficiently contribute to leukocyte classification.

Table 1 summarizes the main methods that use deep learning models found in the literature. The main advantage of deep learning models is that they are highly efficient since they extract the most important information available in the data to generate the prediction. The models presented above have shown an excellent performance higher than 90% [23,35,40], which makes them a helpful tool in the WBC classification. However, their architecture implies a high computational cost due to the use of complex architecture and large volumes of data, this being its main disadvantage. Simpler models that maintain good performance with fewer trainable parameters in their architecture could be generated. However, these methods require image processing techniques for feature extraction, making them slower [45,47]. Therefore, to implement deep learning models in a CAD system, methods should have a reasonable trade-off between performance and complexity without affecting the processing speed.

## 3. Materials and Methods

In this research, a multi-level convolutional neural network (ML_CNN) was developed to detect and classify individual WBC obtained from blood smear images.

### 3.1. White Blood Cell Images Datasets

In this work, five different datasets were used. The description of these sources follows.

*Blood Cell Detection (BCD) dataset* (Aslan [52]): Contains 100 annotated images in png format, with 2237 labeled as Red Blood Cells and only 103 as White Blood Cells. Each image consists of 256 pixels in height and width of RGB channels. (More information can be found at https://github.com/draaslan/blood-cell-detection-dataset. Accessed date: 30 June 2020)*Complete Blood Count (CBC) dataset* (Alam et al. [53]): Contains 360 blood smear images along with their annotation files. (More information can be found at [54], https://github.com/MahmudulAlam/Complete-Blood-Cell-Count-Dataset. Accessed date: 20 June 2021)*White Blood Cells (WBC) dataset* (Zheng et al. [55,56]): Contains 300 images of size 120×120, and 100 color images of size 300×300. (More information can be found at http://www.doi.org/10.17632/w7cvnmn4c5.1. Accessed date: 15 May 2019)*Kaggle Blood Cell Images (KBC) dataset* (Mooney [57]): Contains 12,500 augmented images of blood cells (JPEG) with accompanying cell type labels (CSV). There are approximately 3000 images for each of 4 different cell types. (More information can be found at https://www.kaggle.com/paultimothymooney/blood-cells. Accessed date: 15 May 2019)*Leukocyte Images for Segmentation and Classification (LISC) dataset* (Rezatofighi et al. [58]): Corresponds to a dataset of 250 blood smear images in BMP format. It contains a set of 25 basophil images. (More information can be found at http://users.cecs.anu.edu.au/~hrezatofighi/Data/Leukocyte%20Data.htm. Accessed date: 15 May 2019)

In the previous datasets, there was an insufficient number of basophils. Therefore, the algorithm was developed using images that contain monocytes, lymphocytes, segmented, and eosinophils.

For developing the proposal, a human specialist used the LabelImg (https://github.com/tzutalin/labelImg, Accessed date: 10 September 2019) graphical annotation tool to label the blood smear images. For the first level, a subset of images obtained from the *KBC* data set was selected, the cells were detected with the identification of the region of interest (ROI) using bounding boxes, and the cells were labeled as polymorphonuclear or mononuclear. A total of 365 labeled images with the bounding boxes of the cells were used to train the Faster R-CNN of the first level. A human specialist used the LabelImg tool to label the cell images into lymphocytes, monocytes, segmented, and eosinophils. On the one hand, a CNN model was trained with the mononuclear cells to classify between the *lymphocytes* and *monocytes*, and a total of 2282 and 2134 images were used, respectively. On the other hand, a CNN model was trained with the polymorphonuclear cells to classify between the *segmented neutrophils* and *eosinophils*, and a total of 2416 and 2477 images were used, respectively.

### 3.2. A Multi-Level Convolutional Neural Network Approach

For WBC classification, a multi-level and hybrid scheme is proposed, in which the first step corresponds to the detection and separation of leukocytes into mononuclear and polymorphonuclear. Subsequently, the classification of the subtypes in the second level, monocytes, lymphocytes, eosinophils, and neutrophils, is performed, as shown in Figure 1.

A Faster R-CNN allows individual white blood cells to be detected and extracted in the first level. This model is an object detection system that improves on Fast R-CNN by utilizing a region proposal network (RPN) with the CNN model [59]. Thus, this network is structured in two distinct modules: a CNN that proposes regions and a Fast R-CNN detector that uses the proposed regions (Figure 2).

In the Faster R-CNN, the RPN from an image generates a set of proposed rectangular objects, each of which has an objectivity score. These proposals are developed by sliding a small network over the feature map emitted by the last shared convolutional layer, considering an n×n spatial window mapped to a lower-dimensional feature. Multiple region proposals are simultaneously predicted at each sliding window location, where the number of maximum possible suggestions for each area is denoted as *k*. Each of the *k* proposals is parameterized relative to *k* reference boxes, marked as anchors, and associated with a specific scale and aspect ratio. Each anchor is assigned a binary class (an object or not). The positive or negative labels depend on the most significant overlap of intersection over union values to minimize the objective function. Finally, the Fast R-CNN network inputs the evaluated image and the set of object proposals already obtained [60]. First, the whole image is processed with several convolutional and maxima clustering layers to produce a feature map. Then, for each object proposal, a region of interest (RoI) clustering layer extracts a fixed-length feature vector from the feature map. Each feature vector is fed into a sequence of fully connected layers that ultimately branch into two sister output layers: one that produces softmax probability estimates over *K* object classes plus a general “background” type. Another layer makes four real-valued numbers for each of the *K* object classes. Each set of 4 values encodes the refined positions of the enclosures for one of the *K* classes.

The *softmax* function takes an input vector $z=[z_{1},\ldots ,z_{K}]$ and normalizes it into a probability distribution for the *K* classes:(1)$$\sigma {\left(z\right)}_{i}=\frac{e^{z_{i}}}{\sum_{k=1}^{K}e^{z_{k}}}$$

In the proposed model, the Faster R-CNN is trained to detect leukocytes by separating these cells into two distinct classes according to the morphology of their nuclei, creating two labels at this stage:Mononuclear (MN), whose nuclei show morphological unity, and includes lymphocytes and monocytes.Polymorphonuclear (PMN), whose nuclei are divided, and includes segmented neutrophils and eosinophils.

After the first level, the dataset was separated into two cell groups according to the segmentation of the nucleus. In the second level of the proposal, two CNNs were developed: one for classifying mononuclear cells into lymphocytes and monocytes and another for separating polymorphonuclear cells into segmented neutrophils and eosinophils. The CNNs implemented have the MobileNet architecture as a base model with the application of transfer learning, where the weights were pretrained for the ImageNet classification. The output layer was discarded and replaced with a new fully connected classifier.

The MobileNet uses a depthwise separable convolution that reduces the number of parameters compared to conventional CNN. The depthwise separable convolutions factorize the convolution into a depthwise (*dw*) and a pointwise (*pw*) (see Figure 3). On the one hand, the *dw* convolution applied a single filter to each input channel. On the other hand, the *pw* convolution applied a 1×1 convolution to combine the outputs of the *pw* convolution.

The filtered matrix G(x,y) is obtained by applying the classical convolution between the kernel ω and the input matrix F(x,y) with the following equation:(2)$$G\left(x,y\right)=\omega *F\left(x,y\right)=\left(\sum_{\delta_{x}=-k_{i}}^{k_{i}}\sum_{\delta_{y}=k_{j}}^{k_{j}}\omega \left(\delta_{x},\delta_{y}\right)\cdot F\left(x+\delta_{x},y+\delta_{y}\right)\right)+\omega_{bias}$$
where $\omega_{bias}$ is the bias introduced to the convolution product.

Table 2 schematically shows the architecture of the MobileNet. In this article, we set the hyperparameters of the width multiplier α=0.5 and the resolution multiplier ρ=1. However, the width multiplier reduces the input and output channels by half in this work. The first layer uses the classical convolution. The following layers are depthwise separable convolutions followed by batch-normalization and *ReLU* activation functions. The ReLU activation function has the following equation:(3)ReLu(x)=max{0,x}

The stride is a parameter that controls the movement of the filter over the image and induces the kernels’ downsampling. For instance, a stride of 2 (s2) reduces the image to half the original size. A final average pooling reduces the spatial resolution to 1 before the fully connected layer.

The model was trained using the backpropagation algorithm with the Adam optimizer and the binary cross-entropy loss function:(4)$$L_{cross\_entropy}\left(y,\hat{y}\right)=-\frac{1}{N}\sum_{i=1}^{N}\left[y_{i}\log\left({\hat{y}}_{i}\right)+\left(1-y_{i}\right)\log\left(1-{\hat{y}}_{i}\right)\right]$$
where $y_{i}$ and ${\hat{y}}_{i}$ are the *i*-th target and the predictions, respectively. *N* is the total number of samples.

The hyper-parameters of the learning algorithm were set to learning_rate=0.001, beta_1=0.9, beta_2=0.999, and $epsilon=10^{-7}$.

In this work, we have used a web-based virtual experimental environment. We have run the models using the Kaggle and Google Colab environment with GPU, and the models were implemented with Keras and TensorFlow.

### 3.3. Performance Metrics

The confusion matrix corresponds to a summary of the prediction results obtained with the machine learning model. Given *n* samples, the TP is the number of true positives; the TN is the number of true negatives; FN is the number of false negatives; and FP is the number of false positives.

We use the classification metrics obtained from the confusion matrix to evaluate the performance. These metrics are accuracy, precision, recall, and F1-score, and they are described below.

Accuracy: the accuracy value refers to how close a measurement is to the true value, and the equation is given by
(5)$$Accuracy=\left(\frac{TP+TN}{TP+TN+FP+FN}\right)\cdot 100\%$$
where TN are true negatives.Recall: the measure of sensitivity or recall is the percentage of positive cases that were correctly labeled by the model. The recall equation is given by
(6)$$recall=\left(\frac{TP}{TP+FN}\right)\cdot 100\%$$
where TP is the ratio of true positives, FP is the ratio of false positives, and FN are the false negatives.Precision: precision is the percentage of correct classifications. This metric is defined with the following equation:
(7)$$\mathrm{precision}=\left(\frac{TP}{TP+FP}\right)\cdot 100\%$$F_Score: F_score corresponds to the harmonic mean between precision and sensitivity and gives a trade-off measure between the recall and the precision:
(8)$$F\_Score=2\left(\frac{\mathrm{precision}\cdot Recall}{\mathrm{precision}+Recall}\right)$$

## 4. Results and Discussion

As mentioned in the previous sections, a two-level hybrid model was run to classify white blood cells in blood smears. To evaluate the performance of our proposed model, we have used Monte Carlo cross-validation with 10 repetitions, with a split of 70% for training and 30% for testing. The averages and standard deviations of the metrics computed for the test set are shown in Table 3. It can be seen that a good performance was obtained, with metrics from 96% reaching up to 100%. All the performance metrics were around 98.4%, highlighting the model’s discriminatory power developed for classifying leukocytes.

Table 4 shows a comparative analysis of the results with those obtained in the state of the art. Our proposed model achieved higher performance than those reported by Abou et al. [33], Baydilli [44], Banik et al [47], Huang et al. [39], Jiang et al. [37], Kutlu et al. [49], Liang et al. [50], Özyurt [42], Patil et al. [43], Togacar et al. [34], Wang et al. [35], Yao et al. [38], and Yu et al. [51], who reported accuracy between 83% and 98%. However, it should be noted that the average performance of our proposal was lower than those reported by Baghel et al. [45] and Basnet et al. [36], where they have included image processing for feature extraction to enhance the prediction performance. Likewise, the works of Çınar et al. [23], Hedge et al. [48], and Khan et al. [40] have reported accuracy values higher than 99%. Nevertheless, it should be noted that the latter models have more complex structures and a larger number of trainable parameters, which represents a disadvantage in terms of computational cost.

It is essential to highlight some advantages in terms of the functionality of our proposal. The first one is that the ML-CNN is a method that involves simpler models and obtains comparable performances. This fact differentiates it, for example, from Wang’s proposal [35], which requires microscopy hyperspectral images and the architecture of a 3D residual block used in the deep hyper model. Likewise, Khan et al. [40] presents a more complex model involving convolutional features followed by a selection strategy to identify cellular subtypes. Alternatively, Yao et al. [38] propose a model based on two modules involving transfer learning. All these models are more complex than our proposal. Therefore, the multi-level structure presented is simpler to implement without affecting the classification performance on a large scale.

Another noteworthy aspect of the functionality of the proposed model is that the multi-level strategy allows for the first phase of cell detection. It is possible to identify the regions of interest in the images to extract the white blood cells to be subsequently classified. With the exception of [41,49], none of the reviewed works focuses on this detection scheme. It should be noted that the proposed multi-level scheme not only allows efficient cell element classification but also reduces processing times by running CNN networks in parallel during stage two. This hybrid scheme is one of the differentiating aspects of the proposal compared to the methods discussed in the state of the art, which are primarily single-level. This makes our proposal functionally efficient for use in automated equipment as a CAD system.

Figure 4 and Figure 5 are examples of the operation of the ML-CNN for mononuclear and polymorphonuclear cells, respectively, showing in both cases examples of correctly and erroneously classified images. The excellent classification performance obtained for the mononuclear may be due to the morphological characteristics being well-differentiated between monocytes and lymphocytes, where the former has an irregularly shaped nucleus, and the latter has a wholly rounded shape [1]. If a partial comparison is made, it can be observed that the hybrid model developed obtained a better mononuclear classification accuracy compared to the proposals of Wang et al. [35], Baydilli and Atila [44], and Huang et al. [39]. Although it is ideal to have a model that efficiently achieves a global classification of leukocytes, a model that classifies mononuclear cells very well could help screen viral infections since the count of these subclasses acquires greater relevance in these infections [2].

In the case of polymorphonuclear cells (Figure 5), a good classification of the cellular subtypes was also observed. However, in this group, the identification is more complex since the cellular characteristics are not as well differentiated as in the case of mononuclear cells. In this case, the differentiating aspect is not the nucleus but the cell cytoplasm. Therefore, the staining of the blood smear and the acquisition of the image for its correct identification play a fundamental role. Another aspect that could contribute to misclassification is the shape of the cell nucleus. Although mononuclear cells have a rounded core, in the case of polymorphonuclear cells, a slight indentation generates a lobule shape that is not always constant.

Another aspect to highlight is the image labeling process for the model developed. A medical technologist with extensive professional experience performed the manual labeling, thus minimizing the risk of using erroneous datasets. Likewise, we worked with a subset of random images from four databases that were subsequently labeled for the validation set. Thus, we have a new database whose labels are verified by a professional and that in the future will be available for use in other investigations.

A limitation of the model developed is that, on the one hand, basophils were not considered among the polymorphonuclears due to the small number of cases available in the databases. This issue is a challenge to be faced in future work since basophils are part of a routine blood smear but are found in a shallow frequency, which prevents having enough data to run the training of a machine learning model. It should be noted that in the first instance, it was considered to use a generative network to increase the basophil data. Still, the available cases were scarce; therefore, it was decided not to do so in order not to bias the sample. On the other hand, it is necessary to remember that the proposed model was developed to run a classification of mature leukocytes in blood smears, so it does not include the identification of immature cells that could lead to some pathology such as leukemia. The model would have to be extended to identify these cell subtypes. We intend to include more images for working with this cellular subclass in the classification model for future research.

At the particular point of the images present in the blood smear, which is the type of examination on which this work focuses, we have the additional difficulty that each particulate component of the blood has its shapes, characteristics, internal arrangement, and even color, which are relevant for the classification. Of these components, leukocytes (white blood cells), due to their structural complexity, represent a problem when developing algorithms that have a good level of precision in their classification and detection of each of the cell types that make up this group (lymphocytes, monocytes, eosinophils, segmented, and basophils).

## 5. Conclusions

This paper presents a hybrid multi-level approach for automatically detecting and classifying white blood cells into mononuclear (lymphocytes and monocytes) and polymorphonuclear (segmented neutrophils and eosinophils) types from blood smear images. At the first level, a Faster R-CNN network is applied to identify the region of interest of white blood cells, together with the separation of mononuclear cells from polymorphonuclear cells. Once separated, two parallel convolutional neural networks with the MobileNet structure are used to recognize the subclasses. The model achieved good performance metrics, achieving an average accuracy, precision, recall, and F-score of 98.4%, indicating that the proposal represents an excellent tool for clinical and diagnostic laboratories. Moreover, the proposed ML-CNN approach allows obtaining better accuracy results while optimizing the cost of computational resources, thus allowing the creation, evaluation, and retraining of each neural algorithm in an isolated way, without affecting those that achieve the expected levels of performance.

For further work, on the one hand, we intend to use data augmentation tools to include basophil images in the polymorphonuclear group in training and extend the model for the classification of immature leukocytes. On the other hand, it is also intended to develop machine learning techniques that include expert knowledge to improve performance [61,62].

## Figures and Tables

**Figure 1 diagnostics-12-00248-f001:**
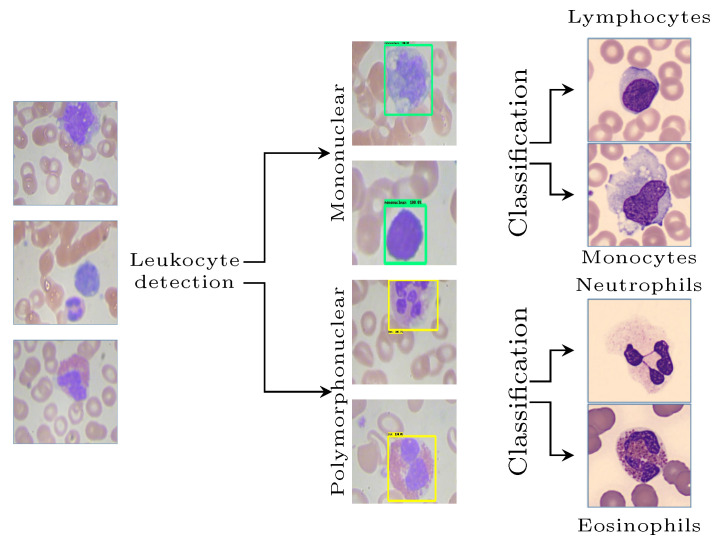
Scheme of identification and classification of white blood cells by the proposed method.

**Figure 2 diagnostics-12-00248-f002:**
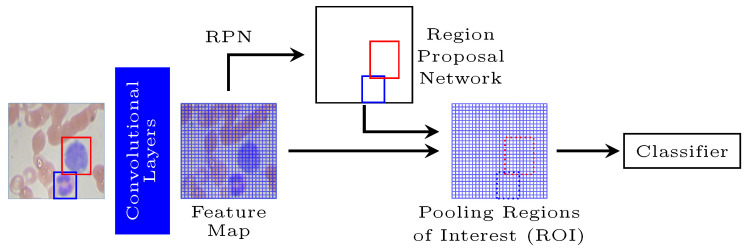
Representation of Faster R-CNN segmentation.

**Figure 3 diagnostics-12-00248-f003:**
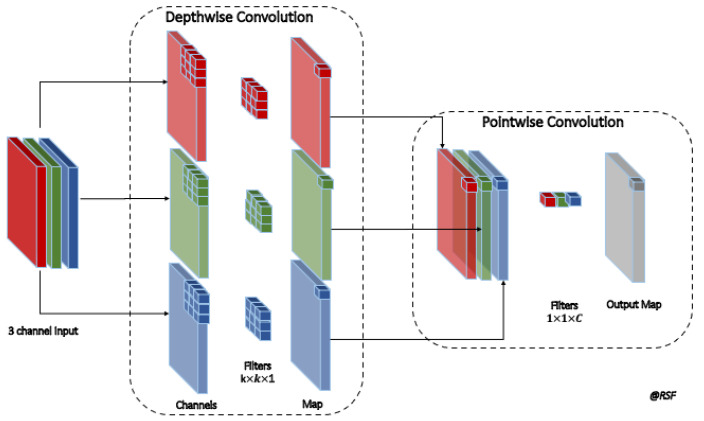
Depthwise separable convolution of the MobileNet, which factorizes the convolution into depthwise and pointwise convolutions.

**Figure 4 diagnostics-12-00248-f004:**
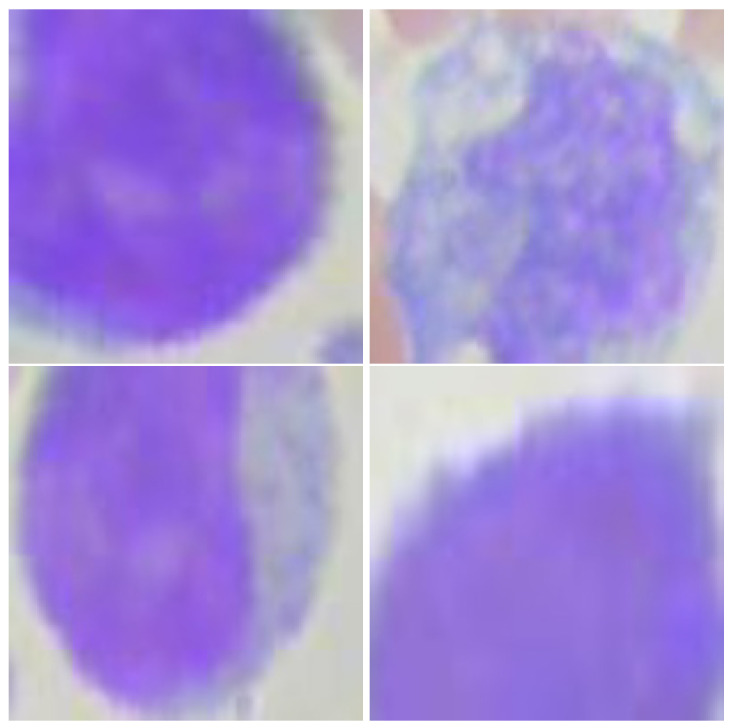
Mononuclear cells classified by the proposed multi-level convolutional neural network. (**upper-left**) Lymphocytes correctly classified; (**upper-right**) monocytes Correctly classified; (**lower-left**) lymphocytes incorrectly classified as monocytes; (**lower-right**) monocytes incorrectly classified as lymphocytes.

**Figure 5 diagnostics-12-00248-f005:**
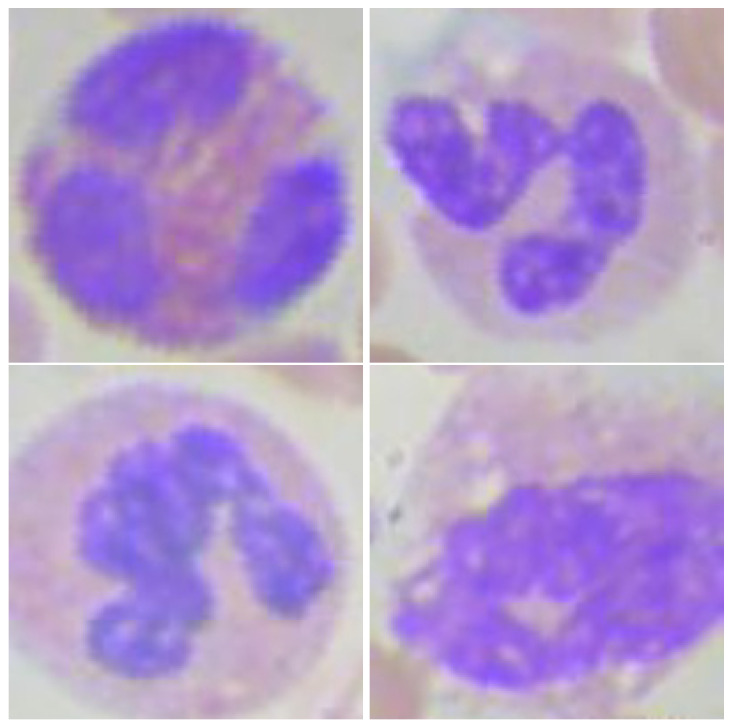
Polymorphonuclear cells classified by the proposed multi-level convolutional neural network. (**upper-left**) Eosinophils correctly classified; (**upper-right**) neutrophils correctly classified; (**lower-left**) eosinophils incorrectly classified as neutrophils; (**lower-right**) neutrophils incorrectly classified as eosinophils.

**Table 1 diagnostics-12-00248-t001:** Summary of the state-of-the-art models for white blood cells classification.

Authors	Model Description
Abou et al. [33]	CNN model with ad hoc structure.
Baghel et al. [45]	CNN model.
Banik et al. [47]	CNN with fusing features in the first and last convolutional layer.
Basnet et al. [36]	DCNN model with image pre-processing and a modified loss function.
Baydilli et al. [44]	WBC classification using a small dataset via capsule networks.
Çınar et al. [23]	Hybrid AlexNet, GoogleNet networks, and support vector machine.
Hegde et al. [48]	AlexNet and CNN model with ad hoc structure.
Huang et al. [39]	MFCNN CNN with hyperspectral imaging with modulated Gabor wavelets.
Jiang et al. [37]	Residual convolution architecture.
Khan et al. [40]	AlexNet model with feature selection strategy and extreme learning machine (ELM).
Kutlu et al. [49]	Regional CNN with a Resnet50.
Liang et al. [50]	Combining Xception-LSTM.
Özyurt [42]	Ensemble of CNN models (AlexNet, VGG16, GoogleNet, ResNet) for feature extraction combined with the MRMR feature selection algorithm and ELM classifier.
Patil et al. [43]	Combining canonical correlation analysis CCANet and convolutional neural networks (Inception V3, VGG16, ResNet50, Xception) with recursive neural network (LSTM).
Razzak [41]	CNN combined with extreme learning machine (ELM).
Togacar et al. [34]	AlexNet with QDA.
Wang et al. [35]	Three-dimensional attention networks for hyperspectral images.
Yao et al. [38]	Two-module weighted optimized deformable convolutional neural networks.
Yu et al. [51]	Ensemble of CNN (Inception V3, Xception, VGG19, VGG16, ResNet50).
ML-CNN (Our proposal)	Multi-level convolutional neural network approach with multi-source datasets. Combines Faster R-CNN for cell detection with a MobileNet for type classification.

**Table 2 diagnostics-12-00248-t002:** Architecture of the MobileNet with transfer learning.

	Layer	Layer Type		Stride	Kernel Size	Input Size	N°Parameters
MobileNet Base Model	1	Conv. 2D		s2	3×3×3×16	128×128×3	496
	2	Conv. dw		s1	3×3×16	64×64×16	208
	3	Conv. pw		s1	1×1×16×32	64×64×16	640
	4	Conv. dw		s2	3×3×32	64×64×32	416
	5	Conv. pw		s1	1×1×32×64	32×32×32	2304
	6	Conv. dw		s1	3×3×64	32×32×64	832
	7	Conv. pw		s1	1×1×64×64	32×32×64	4352
	8	Conv. dw		s2	3×3×64	32×32×64	832
	9	Conv. pw		s1	1×1×64×128	16×16×64	8704
	10	Conv. dw		s1	3×3×128	16×16×128	1664
	11	Conv. pw		s1	1×1×128×128	16×16×128	16,896
	12	Conv. dw		s2	3×3×128	16×16×128	1664
	13	Conv. pw		s1	1×1×128×256	8×8×128	33,792
	14–23	5×	Conv. dw	s1	3×3×256	8×8×256	5×3328
			Conv. pw	s1	1×1×256×256	8×8×256	5×66,560
	24	Conv. dw		s2	3×3×256	8×8×256	3328
	25	Conv. pw		s1	1×1×256×512	4×4×256	133,120
	26	Conv. dw		s1	3×3×512	4×4×512	6656
	27	Conv. pw		s1	1×1×512×512	4×4×512	264,192
Dense	–	Global Avg. Pool		s1	Pool 4×4	4×4×512	-
	28	FC		–	–	512	262,656
	–	Softmax		–	Output	2	1026
Total Parameters: 1,093,218							
Trainable Parameters: 263,682							

**Table 3 diagnostics-12-00248-t003:** Performance obtained in the classification model for each of the WBC cell types considered in the validation set.

Cells	Classification Model	Accuracy	Recall	Precision	F_Score
Mononuclear	Lymphocytes	99.92%±0.08	99.94%±0.08	99.91%±0.08	99.93%±0.07
	Monocytes	99.92%±0.08	99.91%±0.09	99.94%±0.09	99.92%±0.08
Polymorphonuclear	Eosinophils	96.80%±0.30	96.86%±1.17	96.85%±0.61	96.85%±0.32
	Segmented Neutrophils	96.80%±0.30	96.75%±0.70	96.78%±1.14	96.76%±0.27
	Average	98.36%	98.37%	98.37%	98.36%

**Table 4 diagnostics-12-00248-t004:** Comparison of WBC classification results with models in the sate of the art. (NI denotes not informed.)

Authors	Accuracy (%)	Recall (%)	F Score(%)	Layers	Parameters
Abou et al. [33]	96.8	NI	NI	5	NI
Baghel et al. [45]	98.9	97.7	97.6	7	519,860
Baydilli et al. [44]	96.9	92.5	92.3	6	8,238,608
Banik et al. [47]	97.9	98.6	97.0	10	$10^{5}$
Basnet et al. [36]	98.9	97.8	97.7	4	NI
Çınar et al. [23]	99.7	99	99	8 22	$60\cdot 10^{6}$ (AlexNet) $7\cdot 10^{6}$ (GoogleNet)
Hegde et al. [48]	98.7	99	99	8	$60\cdot 10^{6}$ (AlexNet)
Huang et al. [39]	97.7	NI	NI	4	NI
Jiang et al. [37]	83.0	NI	NI	33	NI
Khan et al. [40]	99.1	99	99	8 3	$60\cdot 10^{6}$ (AlexNet) $40\cdot 10^{6}$ (ELM)
Kutlu et al. [49]	97	99	98	50	$26\cdot 10^{6}$ (Resnet50)
Liang et al. [50]	95.4	96.9	94	71	$23\cdot 10^{6}$ (Xception)
Özyurt [42]	96.03	NI	NI	8 22 16 50	$60\cdot 10^{6}$ (AlexNet) $7\cdot 10^{6}$ (GoogleNet) $138\cdot 10^{6}$ (VGG16) $26\cdot 10^{6}$ (Resnet)
Patil et al. [43]	95.9	95.8	95.8	71	$23\cdot 10^{6}$ (Xception)
Razzak et al. [41]	98.8	95.9	96.4	3	NI
Togacar et al. [34]	97.8	95.7	95.6	8	$60\cdot 10^{6}$ (AlexNet)
Wang et al. [35]	97.7	NI	NI	18	$30\cdot 10^{6}$
Yao et al. [38]	95.7	95.7	95.7	55	$60\cdot 10^{6}$
Yu et al. [51]	90.5	92.4	86.6	48 71 19 50	$23\cdot 10^{6}$ (InceptionV3) $23\cdot 10^{6}$ (Xception) $138\cdot 10^{6}$ (VGG19) $26\cdot 10^{6}$ (Resnet50)
ML-CNN (Our proposal)	98.4	98.4	98.4	28	$1\cdot 10^{6}$ (MobileNet)

## Data Availability

The data that support the findings of this study are openly available in a public repositories.

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
