# Peer review of "An Efficient Multi-Level Convolutional Neural Network Approach for White Blood Cells Classification"

_diagnostics, 2022, doi:10.3390/diagnostics12020248_

Round 1

Reviewer 1 Report

There are some suggestions for revision.

1. Two listed contributions are a little bit weak. Please highlight the innovations of the proposed solution.

2. As shown in Section 2, the pros and cons of existing solutions should be discussed.

3. More technical details of the proposed solution should be given, such as equations and mathematical analysis.

4. Please add the related references for the datasets used in this paper.

5. What is the experimental environment?

6. As shown in Tab. 2, is it possible to introduce more objective evaluation indicators, such as recall, precision, and F_score?

7. Figs. 4-7 are similar. Is it possible to show 1-2 figures with obvious differences? 

Author Response

The authors appreciate the time spent by the reviewers in order to provide comments and suggestions that allow us to improve our work.  In what follows, we give answers to each of the given observations of the reviewers. 

1. Two listed contributions are a little bit weak. Please highlight the innovations of the proposed solution.

We appreciate the observation. We have better highlighted the innovations of our proposed solution. See lines 90-101. 

“The ML-CNN proposal was implemented using the MobileNet architecture as the base model. It is an efficient model with an adequate balance between high performance and structural complexity, making its implementation in automated equipment such as a CAD system feasible. Furthermore, the MobileNet applies the depthwise separable convolution to extract relevant features from each channel, which better uses the information contained in the images to improve leukocyte classification.”

2. As shown in Section 2, the pros and cons of existing solutions should be discussed.

Thanks for the suggestion and observation. We have improved the discussion of the pros and cons in section 2 between lines 155-166. Moreover, we have included table 1 that summarizes the main deep learning methods found in the literature.

3. More technical details of the proposed solution should be given, such as equations and mathematical analysis.

Thanks for the suggestion and observation. We have strengthened the article with a mathematical description of the methods. The article was modified accordingly between lines 152-280, and we have included Table 3 for more details of the architecture of the CNN. 

4. Please add the related references for the datasets used in this paper.

Thanks for the suggestion. We have included the links and the papers of the datasets used in this paper. See section 3.1.

5. What is the experimental environment?

We appreciate the observation. In this work, we have used a web-based virtual experimental environment. We have run the models using the Kaggle and Google Colab environment with GPU, and the models were implemented with Keras and TensorFlow. See line 281-283

6. As shown in Tab. 2, is it possible to introduce more objective evaluation indicators, such as recall, precision, and F_score?

Thanks for the comment and observation. We have introduced more performance metrics in Table 4. The comparative analysis is given between lines 306- 317.

7. Figs. 4-7 are similar. Is it possible to show 1-2 figures with obvious differences? 

We appreciate the suggestion. We have reduced figures 4 to 7 into two figures showing only one representative case for each situation. See the new figures 4 and 5. 

Reviewer 2 Report

This paper can be accepted after technique improving in accuary comparing to the state-of-arts.

Author Response

The authors appreciate the time spent by the reviewers in order to provide comments and suggestions that allow us to improve our work.  In what follows, we give answers to each of the given observations of the reviewers.

1. This paper can be accepted after technique improving in accuracy comparing to the state-of-arts.

We appreciate the observation. We have improved the CNN model with a MobileNet and we obtained better results than most of those given in the state-of-art. Our model has an average performance of 98.4% but with a simpler structure than those proposed in the state-of-art.

Round 2

Reviewer 1 Report

All my concerns have been addressed. I recommend this paper for publication.